# Movement Protein Mediates Systemic Necrosis in Tomato Plants with Infection of Tomato Mosaic Virus

**DOI:** 10.3390/v15010157

**Published:** 2023-01-04

**Authors:** Qiansheng Liao, Ge Guo, Ran Lu, Xiaoyi Wang, Zhiyou Du

**Affiliations:** College of Life Sciences and Medicine, Zhejiang Sci-Tech University, Hangzhou 310018, China

**Keywords:** tomato mosaic virus, *Solanum lycopersicum*, systemic necrosis, movement protein, RNA silencing suppressor

## Abstract

The necrogenic strain N5 of tomato mosaic virus (ToMV-N5) causes systemic necrosis in tomato cultivar Hezuo903. In this work, we mapped the viral determinant responsible for the induction of systemic necrosis. By exchanging viral genes between N5 and a non-necrogenic strain S1, we found that movement protein (MP) was the determinant for the differential symptoms caused by both strains. Compared with S1 MP, N5 MP had an additional ability to increase virus accumulation, which was not due to its functions in viral cell-to-cell movement. Actually, N5 MP, but not S1 MP, was a weak RNA silencing suppressor, which assisted viral accumulation. Sequence alignment showed that both MPs differed by only three amino acid residues. Experiments with viruses having mutated MPs indicated that the residue isoleucine at position 170 in MP was the key site for MP to increase virus accumulation, but also was required for MP to induce systemic necrosis in virus-infected tomato plants. Collectively, the lethal necrosis caused by N5 is dependent on its MP protein that enhances virus accumulation via its RNA silencing suppressor activity, probably leading to systemic necrosis responses in tomato plants.

## 1. Introduction

Tomato (*Solanum lycopersicum* L.) is one of the most economically important vegetable crops in the whole world. Unfortunately, many tomato cultivars are susceptible to a number of plant viruses and viroids, which cause substantial yield loss and quality deterioration. Tomato is thought to be one of the most popular plants with infections of plant viruses since over 300 virus species have been determined in tomato plants so far [1].

Tomato mosaic virus (ToMV) is an economically important tomato virus, belonging to the genus *tobamovirus* that includes other tomato viruses, such as tobacco mosaic virus (TMV), tomato brown rugose fruit virus (ToBRFV) and cucumber green mottle mosaic virus (CGMMV) [2,3,4]. ToMV has a positive-sense, single-strand genomic RNA (+ssRNA), which encodes five functional proteins. The 126 kDa protein is one of the viral replicases, which is produced from the translation of viral genome. The 183 kDa protein is the viral RNA dependent RNA polymerase (RdRP), which is the read-through product of the 126 kDa open reading frame (ORF). Both 126 kDa and 183 kDa proteins work together to recruit host factors and replicate viral RNAs [5]. Both 30 kDa protein (movement protein, MP) and 17.5 kDa protein (coat protein, CP) are translated from two individual subgenomic RNAs [6,7]. In addition, both ToMV and TMV encode an extra ORF (ORF6) that spans from the 3′ terminus of MP to the 5′ terminus of CP and produces an approximate 4 kDa protein associated with viral pathogenicity [8,9,10]. Out of all tobamoviral proteins, only 126 kDa protein has been characterized to be an RNA silencing suppressor (RSS) to fight host small-interfering RNA (siRNA)-mediated antiviral defense [11,12,13,14,15]. Interestingly, Vogler and colleagues [16] reported that TMV MP contributes to host antiviral defense by promoting the intercellular spread of antiviral silencing signals in host plants.

All viral proteins encoded by tobamoviruses have been shown to be important for the development of viral symptoms [3,6,8,17]. For instance, the small replicase 126 kDa of TMV is responsible for the expression of the mosaic symptoms on tobacco plants [18]. Moreover, the replicase of TMV and other tobamoviruses can trigger tobacco *N* gene-mediated hypersensitive response (HR), leading to the development of local lesions in the infected sites [19]. TMV CP protein also contributes to the induction of severe yellow mosaic in tobacco [20] and functions as an HR elicitor in *Nicotiana sylvestris* and *Solanum melongena* (eggplant) [21,22]. Similarly, ToMV MP is an avirulence protein, which is recognized specifically by the tomato resistance gene *Tm-2*^2^, leading to extreme resistance in tomatoes [23]. However, such *R* gene-mediated resistance can be broken through mutations of viral avirulence proteins [24,25,26].

Viral symptoms of tomato plants with infection of ToMV can be diverse, depending on tomato cultivars, viral strains, and environmental conditions. In general, tomato plants with ToMV infection show mosaic, yellowing, and mottling symptoms on their leaves. Systemic necrosis is the most severe viral symptom, usually leading to plant death. Systemic necrosis of tomato plants has been reported in some cases of virus infections, including ToMV [24], tomato torrado virus (ToTV) [27], rehmannia mosaic virus (ReMV) [28], parietaria mottle virus (PMoV) [29], and some satellite RNAs of cucumber mosaic virus (CMV) [30,31,32]. The strain N5 of ToMV (ToMV-N5, referred to as N5 afterward) that we isolated previously from the field-grown tomato plants induces systemic necrosis in tomato plants [33,34]. In this study, we determined the MP protein of ToMV-N5 to be responsible for the induction of tomato necrosis. Interestingly, we found that N5 MP is a weak RSS, which increases virus accumulation in both *Nicotiana benthamiana* and tomato plants. Experiments with viruses harboring mutations in MP indicate that the Ile residue at position 170 of MP is essential for its RSS activity, but also is required for the induction of systemic necrosis. This suggests that N5 MP functions as an RSS to increase viral protein levels that trigger systemic necrosis response in tomato plants.

## 2. Materials and Methods

### 2.1. Viruses and Host Plants

Both ToMV strains N5 and S1 were maintained separately in tomato cultivar Zhongshu4 plants. The genome sequences of N5 and S1 have been deposited in the GenBank with the accession numbers GQ280794 and AJ132845, respectively. For growing tomato cultivar Hezuo903, the seeds were kept on wet filter papers at room temperature for 2 days, and then transferred into the pots with soil composts. The seedlings reaching the stage of 1~2 true leaves were used for viral inoculation. *N. benthamiana* seedlings were potted after sowing for about 10 days and grown to the 4~6 leaf stage for viral inoculation or agroinfiltration. Both plants were grown in a greenhouse under the conditions of 25 °C, 16/8 h (light/dark) light cycles, and a light intensity of 100 μmol/m^2^ and about 60% humidity.

### 2.2. Plasmid Constructs

The plasmids pCB301-N5 and pCB301-S1 are the 35S promoter-derived infectious clones of N5 and S1, respectively, which were constructed through a two-step cloning method. The first step was to amplify the DNA fragment corresponding to the 5′ 3362 nt sequence of each strain, followed by digestion with restriction enzyme *Bam*HI (Thermo Scientific, Vilnius, Lithuania), and insertion into the binary vector pCB301 pre-digested by *Stu*I (Thermo Scientific, Vilnius, Lithuania) and *Bam*HI. The plasmid pCB301 was described previously [35]. The second step was to amplify another DNA fragment corresponding to the remaining 3′ sequence of N5 or S1. After digestion with *Bam*HI and *Sac*I (Thermo Scientific, Vilnius, Lithuania), the DNA fragment was inserted downstream of the first DNA fragment, to create pCB301-N5 or pCB301-S1. pCB301-N5^S1−183KDa^, pCB301-N5^S1-mp^, pCB301-N5^S1-cp^ are the constructs where the coding sequence of183 kDa, MP, or CP in pCB301-N5 were substituted with that of S1 via cloning sites *Ase*I/*Sac*I, *Ase*I/*Bst*EII, or *Bst*EI/*Sac*I, respectively. In the same way, pCB301-S1^N5−183KDa^, pCB301-S1^N5-mp^, pCB301-S1^N5-cp^ were constructed using the same cloning sites. pCB301-N5Δcp and pCB301-S1Δcp are the constructs in which the initiation codon of both *CP* genes was substituted with AGG to prevent CP expression. To express GFP in N5 or S1, a DNA fragment containing the GFP sequence was amplified from the plasmid TMV-gfp [36], digested with *Mlu*I and *Sma*I (Thermo Scientific, Vilnius, Lithuania), and inserted into pCB301-N5 or pCB301-S1, to generate pCB301-N5-gfp or pCB301-S1-gfp. pCB301-N5Δcp-gfp and pCB301-S1Δcp-gfp are the derivatives of pCB301-N5-gfp or pCB301-S1-gfp, respectively, in which the initiation codon of their *CP* gene was substituted with AGG to prevent CP expression. To introduce mutation(s) in S1 MP, DNA fragments harboring point mutation were amplified by overlapping PCR with respective primer pairs and inserted into *Ase*I and *Sma*I-digested pCB301-S1-gfp or *Ase*I-*Bst*EII-digested pCB301-S1 to generate their MP mutants. Using the same procedure, the MP mutants of N5-gfp were constructed.

All the plasmids constructed in this work were confirmed by Sanger DNA sequencing. The primers for generating DNA constructs are listed in Appendix A.

### 2.3. Agrobacterium-Mediated Viral Inoculation

All the plasmids constructed were individually transformed into *Agrobacterium tumefaciens* GV3101 by the freeze–thaw method [37]. The procedures of bacterium cultivation and agroinfiltration were performed as described previously [38]. The final concentration of bacterial cells was set up constantly to 0.5 at OD_595_ for agroinfiltration, with the exception of the assay for testing viral cell-to-cell movement (0.0001 at OD_595_). Viral symptoms on *N. benthamiana* and tomato plants were photographed when desired phenotypes appeared. GFP fluorescence in the leaves of *N. benthamiana* plants was photographed under a long-wavelength UV lamp by a filtered camera (Nikon, Tokyo, Japan).

### 2.4. RNA Extraction and Northern Blot Hybridization

Total RNA was extracted from plant tissue with TRIzol Reagent (Invitrogen, Carlsbad, CA, USA) according to the manufacturer’s instructions. Relative accumulation levels of large molecular weight RNAs, including ToMV RNAs and GFP mRNA, were analyzed using Northern blot. RNA electrophoresis, transfer and hybridization were performed according to the procedure described previously [38]. For analysis of GFP-derived siRNAs (G-siRNAs), 15 µg of RNA sample was loaded into 15% PAGE gel containing 8 M urea for RNA separation, and then electrically transferred to a positively charged Nylon membrane (Amersham, Little Chalfont, UK). G-siRNA was detected using the DIG-labeled DNA oligonucleotides as described previously [39]. The DIG-labeled DNA probes for the detection of ToMV, GFP mRNA, or G-siRNA were shown in Appendix A.

### 2.5. Protein Extraction and Western Blot

Total protein was extracted from 0.1 g of leaf tissue as described previously [40]. Protein samples were separated in a 12% SDS-PAGE, electrically transferred to nitrocellulose membranes (GE, Freiburg, GER), and blotted with GFP antibodies (Santa Cruz, Dallas, TX, USA) as described previously [40]. Western blotting signals were detected with enhanced chemiluminescence solution (Thermo Scientific, Rockford, IL, USA), followed by exposing membranes to X-ray films as described previously [40].

## 3. Results

### 3.1. MP Contributes to the Development of Necrosis Symptoms on the Tomato Plants Infected by ToMV-N5

To determine which viral protein(s) is required for N5 to induce systemic necrosis, here we used another strain, ToMV-S1 (referred as to S1 afterward), as a reference that causes latent infection or mild mosaic on tomato plants [41,42,43]. Firstly, the 35S promoter-based infectious clones of both strains were constructed by inserting the complementary DNA fragment of the viral genome into the binary vector pCB301 and were confirmed to be infectious with the evidence that both ToMV strains caused plant death in *N. benthamiana* plants at 5 days post infiltration (dpi) (Appendix A).

Next, we exchanged the sequence of 183 kDa, MP, or CP between both strains to generate six recombinants (Figure 1a). For instance, the nucleotide sequence of the 183 kDa protein in N5 was replaced with that of S1 to generate the recombinant N5^S1−183KD^, and vice versa. Then, we tested N5 and its derivatives (N5^S1−183KD^, N5^S1-mp^, N5^S1-cp^) as well as S1 on the tomato cultivar Hezuo903. As expected, the plants infected with N5 developed necrosis in the apical leaves, while those infected with S1 just presented mild mosaic and curling leaves at 10 dpi (Figure 1b, upper panel). N5^S1-mp^ harboring the MP gene of S1 did not cause necrosis in the apical leaves, but both N5^S1−183kD^ and N5^S1-cp^ did (Figure 1b, upper panel) as well as the wildtype N5, demonstrating that MP was required for N5 to induce systemic necrosis. Interestingly, N5^S1-mp^ accumulated viral RNAs at a similar level as S1, 62% less than N5 (Figure 1c, upper panel). The reciprocal experiments with S1 derivatives showed that S1^N5-mp^ carrying the MP gene of N5 caused systemic necrosis, as did N5 (Figure 1b, lower panel), and it had a 112% higher accumulation level than S1 (Figure 1c, lower panel). However, both S1^N5−183kD^ and S1^N5-cp^ had weak pathogenicity and accumulated to low levels as did S1 (Figure 1b,c, lower panel). Taken together, these data demonstrate that MP engages in the development of tomato necrosis caused by N5, and also regulates viral titer in the host plants.

### 3.2. N5 MP Had the Ability to Increase Virus Accumulation in N. benthamiana

To better understand the role of MP in viral accumulation, we constructed the GFP-expressing versions of both ToMV strains by inserting the GFP coding sequence downstream of their CP promoter (Figure 2a). Meanwhile, the MP gene in both GFP-expressing strains was exchanged to generate two MP recombinants (Figure 2a). It is worth mentioning that both GFP-expressing ToMV strains and their MP recombinants do not cause plant death in *N. benthamiana* since the GFP insertion destroys ORF6 that is required for induction of *N. benthamiana* death during viral infection [8,9]. At 8 dpi, N5-*gfp* produced strong GFP fluorescence in the top systemic leaves of *N. benthamiana*, while the GFP fluorescence emerged from the S1-*gfp-*infected plants was much weaker (Figure 2b). Replacement of the MP gene in N5-*gfp* with S1 MP reduced GFP fluorescent (N5^S1-mp^-*gfp* vs. N5-*gfp*), and vice versa (S1^N5-mp^-*gfp* vs. S1-*gfp*) (Figure 2b). However, we noticed that S1^N5-mp^-*gfp* was not strong as N5-*gfp*, and N5^S1-mp^-*gfp* was stronger than S1-*gfp* in the production of GFP fluorescence. The differential intensities of GFP fluorescence produced by these four GFP-expressing viruses were supported by RNA gel blotting analyses of viral RNAs and immunoblotting analyses of the GFP protein (Figure 2c). Collectively, our data demonstrate that N5 MP has advantages over S1 MP in assisting viral accumulation in plants and suggest that the differential accumulation of both ToMV strains in plants is largely attributed to their MPs.

### 3.3. N5 MP May Increase Viral Accumulation by Stabilizing Viral RNAs

To determine whether both MPs have differential ability to assist viral accumulation in local leaves, we tested infections of N5-Δcp-*gfp*, S1-Δcp-*gfp*, and their MP variants (N5^S1-mp^-Δcp-*gfp*, S1^N5-mp^-Δcp-*gfp*) (Figure 3a) in *N. benthamiana* via agroinfiltration in the absence of exogenous RSS. At 4 dpi, the patch inoculated with N5-Δcp-*gfp* produced strong GFP fluorescence in the infiltrated area, while the patch inoculated with S1-Δcp-*gfp* showed extremely weak fluorescence (Figure 3b). Both MP variants N5^S1-mp^-Δcp-*gfp* and S1^N5-mp^-Δcp-*gfp* had a similarly strong GFP fluorescence, which was stronger than that of S1-Δcp-*gfp*, but weaker than that of N5-Δcp-*gfp* (Figure 3b). The differential GFP fluorescence observed was consistent with the results of RNA gel blotting analysis of GFP-expressing subgenomic RNA and immunoblotting analysis of GFP protein (Figure 3c). As expected, co-expression of p19 with each variant substantially enhanced the GFP fluorescence (Figure 3d). No discernable difference in GFP phenotype was observed between N5-Δcp-*gfp* and N5^S1-mp^-Δcp-*gfp*, or between S1-Δcp-*gfp* and S1^N5-mp^-Δcp-*gfp* (Figure 3d). This is consistent with the similar accumulation levels of viral subgenomic RNA and the GFP protein between them (Figure 3e). These results indicate that N5 MP has advantages over S1 MP in assisting virus accumulation and suggest that the advantage is associated with the enhanced stability of viral RNAs, rather than increased viral replication.

### 3.4. MPs of Both ToMV Strains Are Equally Efficient in Mediating Viral Cell-to-Cell Movement

The distinct ability of both MPs in assisting viral infections promoted us to determine whether they are differential in facilitating viral cell-to-cell movement in local leaves. To do this, we compared the cell-to-cell movement of N5-Δcp-*gfp* and N5^S1-mp^-Δcp-*gfp* by infiltrating agrobacterium cells at 0.0001 OD_595_ (Figure 4a). In this assay, the RSS p19 was co-expressed with either virus. Both GFP-expressing variants are deficient in systemic movement due to lacking the CP protein. The highly diluted agrobacteria would produce single cell infections, which were separated spatially and could be used to compare viral cell-to-cell movement rate from infected cell to adjacent cells with the help of GFP fluorescence. Green fluorescence spots developed in the same leaf infiltrated with N5-Δcp-*gfp* (left half) or N5^S1-mp^-Δcp-*gfp* (right half) began to be observable under UV light at 4 dpi and became more and larger gradually with time going by 7 dpi (Figure 4b). Measurement of GFP foci from five infiltrated leaves showed that the distribution and mean sizes of GFP foci were almost the same between N5-Δcp-*gfp* and N5^S1-mp^-Δcp-*gfp* at 5, 6, or 7 dpi (Figure 4c). This suggests that both MPs were equally efficient in promoting viral cell-to-cell movement, and N5 MP would have other function(s), rather than facilitation of cell-to-cell movement, to confer higher viral accumulation.

### 3.5. N5 MP Is a Weak RNA Silencing Suppressor

Since N5 MP has the ability to enhance the stability of viral RNA, we asked whether N5 MP has an RSS activity. To do this, we tested the capability of both MP proteins in the inhibition of GFP post-transcriptional gene silencing (PTGS) induced by a hairpin RNA of GFP (dsFP) mimicking viral replication intermediates as did previously [39]. Meanwhile, the well-studied RSS p19 was included as a positive control. As expected, dsFP induced strong GFP silencing in the control leaves (vector) with invisible GFP fluorescence, while the dsFP-induced GFP silencing was inhibited greatly in the presence of p19 as showing strong GFP fluorescence at 4 dpi (Figure 5a). GFP phenotype in the leaves expressing S1 MP was invisible as the vector control. In contrast, the leaves with the expression of N5 MP exhibited GFP fluorescence, which was markedly weaker than that emerged in the leaves expressing p19 (Figure 5a). The differential GFP phenotype was supported by the accumulation of the GFP mRNA and protein in the infiltrated leaves (Figure 5b). RNA blotting analyses of GFP-derived siRNAs (G-siRNAs) showed that p19 dramatically inhibited production of siRNAs, while both S1 MP and N5 MP had no obvious impact on the steady-state level of G-siRNAs, compared with the vector control (Figure 5b). These results support our speculation that N5 MP, but not S1 MP, functions as a weak RNA silencing suppressor to enhance virus levels in host plants.

### 3.6. The Residue at Position 170 of MP Is the Determinant to the Differential Accumulation of Both ToMV Strains in N. benthamiana

To determine the key residue(s) responsible for the differential accumulation of both ToMV strains in host plants, here we aligned the amino acid sequences of N5 MP and S1 MP. The sequence alignment showed that only three residues vary at positions 92, 170, and 240, where N5 MP encodes Met (M), Ile (I), and Tyr (Y), while S1 MP encodes Leu (L), Thr (T) and Asp (D), respectively (Appendix A). Subsequently, single or double mutations were introduced into the MP sequence of S1-*gfp* to generate six mutants, S1^92M^-*gfp*, S1^170I^-*gfp*, S1^240Y^-*gfp*, S1^92M170I^-*gfp*, S1^92M240Y^-*gfp*, and S1^170I240Y^-*gfp*. The upper systemic leaves of *N. benthamiana* plants inoculated with these mutants showed varied GFP fluorescence at 8 dpi (Figure 6a). In these three single mutants, only S1^170I^-*gfp* produced a strong GFP phenotype as did S1^N5-mp^-*gfp*, while the other two were similarly weak to S1-*gfp*. Both double mutants (S1^92M170I^-*gfp*, S1^170I240Y^-*gfp*) containing the residue 170I were similar to the single mutant S1^170I^-*gfp* in producing GFP fluorescence (Figure 6a). To further demonstrate the importance of the residue at position 170, reciprocal mutants were made in N5-*gfp* and tested on *N. benthamiana* as well (Figure 6b). N5-*gfp* mutants harboring 170T in single (N5^170T^-*gfp*) or double mutants (N5^92L170T^-*gfp*, N5^170T240D^-*gfp*) produced similarly weak GFP fluorescence as did N5^S1-mp^-*gfp*, which was much weaker than N5-*gfp* or other mutants (Figure 6b). Taken together, these results indicate that the distinct residues at position 170 in MP determine the differential accumulation of both ToMV strains.

### 3.7. The Key Residue 170I of MP Is Required for the Development of Necrosis Phenotype in Tomato Plants Infected with ToMV

Three residues differ between N5 MP and S1 MP as aforementioned (Appendix A). To explore which residue(s) in N5 MP is pivotal for the induction of necrosis symptoms in tomato plants, these differential residues were individually or doubly introduced into S1 to generate six S1 mutants, including S1^mp92M^, S1^mp170I^, S1^mp240Y^, S1^mp92M170I^, S1^mp92M240Y^, and S1^mp170I240Y^. All these mutants, as well as wildtype, were tested in the tomato cultivar Hezuo903 plants. Interestingly, these mutants S1^mp170I^, S1^mp92M170I^, and S1^mp170I240Y^ that harbor 170I induced necrotic phenotype in the tomato plants at 10 dpi, which is similar to that caused by S1^N5-mp^ (Figure 7a). However, the other three mutants were non-necrotic, just as the wildtype S1 (Figure 7a). This indicates that the residue 170I of MP is critical for MP to induce systemic necrosis in N5-infected tomato plants. RNA blotting analyses showed that these necrosis-inducing mutants accumulated at a similarly high level to S1^N5-mp^, while other non-necrotic mutants accumulated at the level as did S1 (Figure 7b), indicating the relationship between virus titer and necrosis induction.

## 4. Discussion

In this study, we investigated the molecular properties of ToMV-N5 that induces systemic necrosis in tomato plants. Our data indicate that N5 MP is the pathogenic determinant for necrosis development in tomato plants. Compared with the MP protein of the non-necrotic S1 strain, N5 MP has advantages in increasing virus titer, but does not facilitate viral cell-to-cell movement in plants. Interestingly, N5 MP, but not S1 MP, is a weak RNA silencing suppressor, which assists virus accumulation in plants. Experiments with viruses having mutated MPs indicate that the residue 170I is required for N5 MP not only to increase viral accumulation, but also to induce necrotic symptoms in tomato plants. The role of MP in the induction of systemic necrosis is discussed below.

One of the most destructive symptoms caused by plant viruses is systemic necrosis, which usually results in plant death. Several viruses have been reported to cause systemic necrosis in tomato plants [24,27,29], while the underlying basis of inducing such severe disease is largely unknown. The vp26 subunit of viral CP contributes to the occurrence of systemic necrosis in tomato plants infected with ToTV [27]. Mutations of two residues in the 130 kDa replicase of tobacco mild green mosaic virus (TMGMV) assist the virus to escape the *tm-1* resistance gene-mediate inhibition of virus replication, leading to the development of systemic necrosis in virus-infected tomato plants [44]. Here, we determined that N5 MP contributed to the induction of systemic necrosis in the virus-infected tomato plants (Figure 1). Wieczorek and Obrępalska-Stęplowska [45] reported that one residue change in the MP protein of ToTV caused systemic necrosis of tomato due to enhanced viral infection. However, we found that MP of the non-necrotic strain S1 is equally efficient in facilitating viral spread locally as N5 MP, indicating that the N5 MP mediated systemic necrosis of tomato plants is not attributed to its function in facilitating cell-to-cell movement.

How does MP contribute to the induction of systemic necrosis in virus-infected tomato plants? One of the explanations can be the increased virus titer. This explanation can be supported by several lines of experimental data. Firstly, compared with the non-necrotic strain S1, the necrotic strain N5 has an approximately 2.5 times higher viral accumulation in tomato plants (Figure 1). Secondly, replacement of the *MP* gene in S1 with N5 *MP* leads to increased virus titer by 112% and occurrence of systemic necrosis, and vice versa (Figure 1). Third, the residue Ile at position 170 of N5 MP contributes to both virus accumulation and systemic necrosis (Figure 6 and Figure 7). Production of systemic necrosis caused by increasing viral titer has been reported in the Argonaute 2 (AGO2)-silenced tomato plants infected with PVX [46]. Increased viral titer implies more viral proteins produced during infection, which would be responsible for the induction of systemic necrosis. As a well-known RNA silencing suppressor, the 2b protein of cucumber mosaic virus (CMV2b) induces systemic necrosis in *Nicotiana glutinosa* [47,48], and forcing CMV2b into the nucleus causes systemic necrosis of *Arabidopsis thaliana* [38]. Inaba et al. [49] reported that CMV-induced *Arabidopsis* necrosis is the outcome of CMV2b binding to host catalase 3 that is important in removing cellular hydrogen peroxide. Thus, it is possible that N5 MP may interact with tomato proteins that are involved in regulation of systemic necrosis.

The MP protein of tobamovirus is involved in viral cell-to-cell spread and is recognized as an avirulence protein by tomato *Tm-2*^2^ resistance gene to activate hypersensitive reaction or extreme resistance against viral infection [4,50,51]. N5 causes systemic necrosis in the tomato cultivar Hezuo903 that contains two heterozygous alleles, the resistance gene *Tm-2*^2^ and susceptible gene *tm-2*, while it induces only mosaic symptoms on another tomato cultivar Zhongshu4 that harbors homozygous alleles *tm-2*/*tm-2* [52]. This strongly suggests that the systemic necrosis in the ToMV-N5-infected tomato cultivar Hezuo903 is the phenotype of the interaction of N5 MP with *Tm-2*^2^. However, ToMV-S1 does not induce systemic necrosis, indicating the *Tm-2*^2^-mediated resistance is specific to certain strains, which is consistent with the previous report that the changes of both key residues Ser238 and Lys244 in ToMV-2a MP confer the ability to overcome the *Tm-2*^2^*-*mediated resistance in tomato [23]. We found that the residue 170I of N5 MP was required to produce systemic necrosis in tomato plants. It would be interesting to determine whether the residue 170I participates in the recognition of *Tm-2*^2^ to ToMV MP.

In tobamoviruses, only the small repliase, such as ToMV 130 kDa and TMV 126 kDa, is an RSS protein [11,13,53]. Here, we found that N5 MP, but not S1 MP, possesses a weak suppressor activity (Figure 5). To our knowledge, this finding is the first report in all MP proteins of tobamoviruses so far. N5 MP does not reduce the steady-stage level of the hairpin RNA-derived siRNAs (Figure 5), implying that N5 MP inhibits siRNA-mediated local RNA silencing by inactivating siRNAs or targeting the step downstream of siRNA production. The small replicase of tobamoviruses is such a RSS protein that inhibits siRNA methylation, instead of siRNA biogenesis [11,13,15,53]. As viral MP, potato virus X (PVX)-encoded P25 functions as an RSS as well [54]. P25 interacts with argonaute 1 (AGO1), the core component of RNA-induced silencing complex, leading to the late degradation via the proteasome pathway [55]. Like N5 MP, P25 triggers systemic necrosis of *N. benthamiana* with a PVX-associated synergism in a threshold-dependent manner [56], which results from P25-induced endoplasmic reticulum (ER) stress [57]. Thus, we cannot rule out the possibility that N5 MP might trigger ER stress, leading to systemic necrosis of tomato. Our data demonstrate that the residue Ile at position 170 of MP is required for increased virus accumulation, suggesting that this residue is critical for the RSS activity of N5 MP. Moreover, this residue is also essential for N5 MP-induced systemic necrosis in virus-infected tomato plants, strongly suggesting that the RSS activity of N5 MP is coupled with the ability to induce systemic necrosis.

Several MPs of plant viruses have been reported to be RSS [58,59,60,61,62,63,64]. p25, one of the triple gene block proteins encoded by the genus potexvirus, is required for viral cell-to-cell movement, and its mutants that are defective for silencing suppressor lost the capability of viral cell-to-cell movement [58]. Red clover necrotic mosaic virus (RCNMV) MP has RNA silencing suppressor activity, which is dependent on the cell-to-cell spread of the virus [60]. Here, we found that in spite of the presence or absence of RNA silencing suppressor activity, both MPs of N5 and S1 are equally efficient in facilitating viral cell-to-cell movement, suggesting that silencing suppressor activity of N5 MP was not required for viral cell-to-cell spread. Vogler et al. [16] reported that TMV MP promoted the spread of RNA silencing signals and functioned as an attenuator of RNA silencing. It remains to be elucidated whether ToMV MP has the capability of enhancing the spread of RNA silencing signals through plasmodesmata to decrease viral levels in host plants.

In summary, this work genetically mapped the viral protein MP to be the determinant for inducing systemic necrosis in tomato plants infected with the necrogenic strain ToMV N5. N5 MP was characterized to be a weak RSS, which assisted virus accumulation in host plants. The key residue Ile170 is required for N5 MP to increase virus accumulation, leading to the development of systemic necrosis in tomato plants.

## Figures and Tables

**Figure 1 viruses-15-00157-f001:**
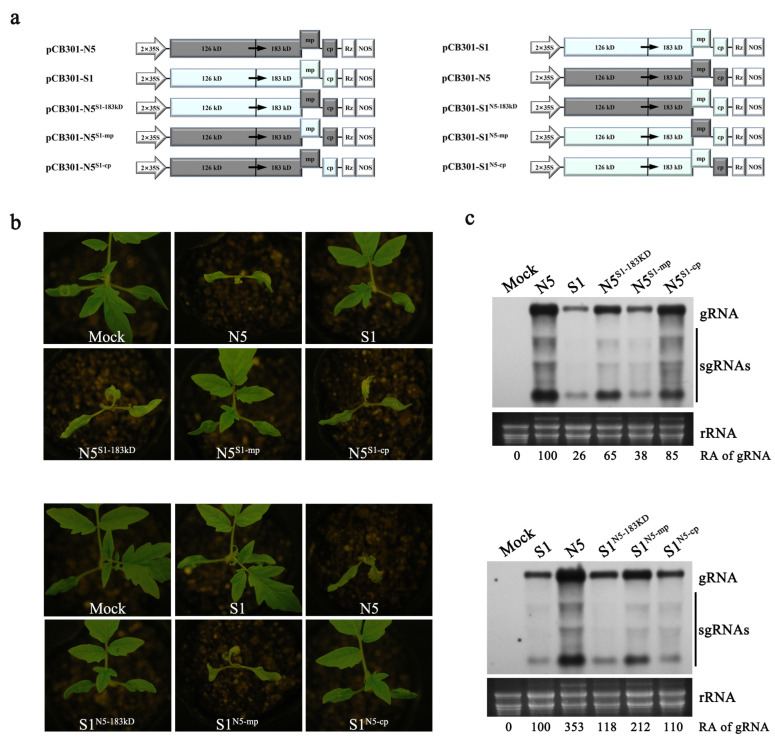
Genetic mapping of the viral protein responsible for the induction of systemic necrosis in tomato plants with infection of ToMV-N5. (**a**) Schematic diagrams of the 35S promoter-based infectious clones of wildtype N5 and S1 strains of ToMV, and their derivatives. The DNA fragments of viral genomes were inserted between the 35S promoter (2 × 35S) and HDV cis-cleaving ribozyme sequence (Rz) in the binary vector pCB301. Viral sequences of N5 and S1 are colored gray and light blue, respectively. Six chimeric viruses expressing heterologous 183 KD protein, MP, or CP were shown. For instance, pCB301-N5S1-183KD is the construct where the 183 KD in N5 was replaced with that of S1, and pCB301-S1N5-183KD is the construct where the 183 KD in S1 was replaced with that of N5. (**b**) Viral symptoms on the tomato plants inoculated with N5, S1, or their chimeric viruses, as illustrated in (**a**). Plants were photographed at 7 days post-infiltration (dpi). (**c**) Northern blotting analyses of viral RNAs in the upper systematic leaves of the infected plants at 7 dpi. Viral RNAs were detected with a digoxigenin (DIG)-labeled DNA oligonucleotide complementary to the sequence of ToMV genomic 3′ untranslated region (3′ UTR). Genomic RNAs and subgenomic RNAs of these viruses were indicated with gRNA and sgRNA, respectively, to the right. Relative accumulation levels of gRNA were shown below. Ethidium bromide (EB)-stained ribosomal RNAs (rRNAs) were used to indicate the relative amounts of RNA samples loaded.

**Figure 2 viruses-15-00157-f002:**
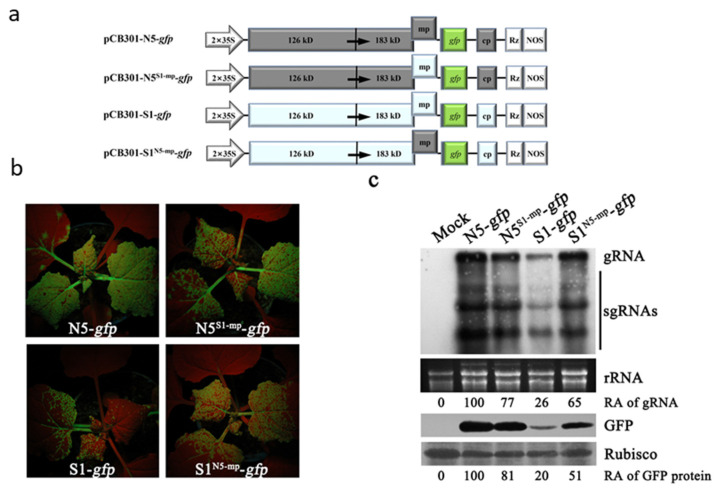
N5 MP increased viral accumulations in *Nicotiana benthamiana.* (**a**) Schematic diagrams of ToMV constructs harboring the coding sequence of green fluorescent protein (GFP). (**b**) Appearance of green fluorescence on the plants inoculated with N5-*gfp*, S1-*gfp*, or their MP recombinants under a long wavelength UV illuminator at 8 dpi. (**c**) Northern blot and immune blot analyses of viral RNAs and GFP proteins in upper systemic leaves, respectively. Genomic RNAs and subgenomic RNAs of these viruses were indicated with gRNA and sgRNA, respectively, to the right. Relative accumulation levels of gRNA and GFP protein are shown below. Ethidium bromide-stained ribosomal RNAs (rRNAs) were used as a loading indicator. The large subunit of Rubisco stained by Ponceau S was used to assess the relative amounts of protein samples loaded.

**Figure 3 viruses-15-00157-f003:**
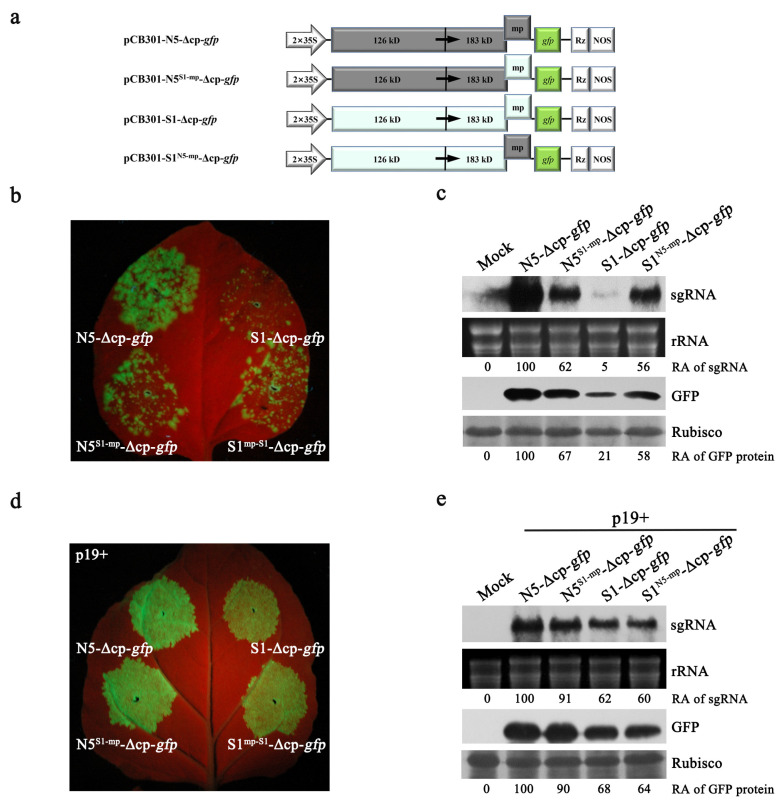
The difference in both N5 and S1 MPs-mediated viral accumulations was restored in the presence of the RNA silencing suppressor p19. (**a**) Schematic diagrams of N5-Δcp-*gfp*, S1-Δcp-*gfp*, and their MP exchangers, where the *CP* gene was deleted. (**b**,**d**) Green fluorescence appearance on the leaves of *N*. *benthamiana* inoculated with N5-Δcp-*gfp*, S1-Δcp-*gfp*, and their MP exchangers in the absence of p19, or in the presence of p19, respectively. GFP fluorescence was photographed at 4 dpi. (**c**,**e**) RNA blotting analyses of GFP-expressing subgenomic RNA (sgRNA) and Western blotting detection of GFP protein from the leaf samples as shown in (**b**,**d**), respectively. Relative accumulation levels of sgRNA and GFP protein are shown below. Ethidium bromide-stained ribosomal RNAs (rRNAs) and Ponceau S-stained Rubisco were used as loading controls for Northern blot and Western blot, respectively.

**Figure 4 viruses-15-00157-f004:**
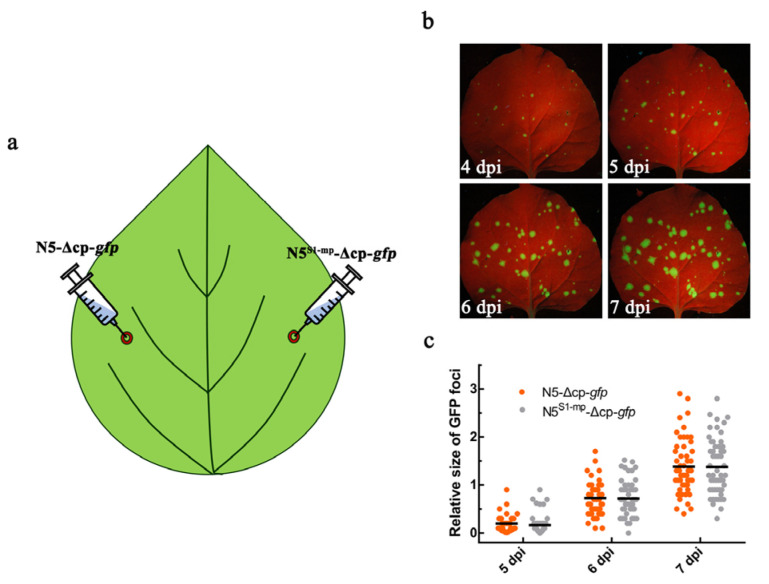
Comparison of both MPs in facilitating viral cell-to-cell movement. (**a**) Schematic diagram of agroinfiltration to inoculate N5-Δcp-*gfp* (left half) and N5^S1-MP^-Δcp-*gfp* (right half) on the same leaf of *N. benthamaina* plants. As shown in Figure 4a, N5-Δcp-*gfp* was constructed by deleting the *CP* gene in N5-*gfp*, and N5^S1-MP^-Δcp-*gfp* was constructed by replacing MP in N5-Δcp-*gfp* with S1-MP. (**b**) Distribution patterns of GFP fluorescence foci produced from N5-Δcp-*gfp* and N5^S1-MP^-Δcp-*gfp* at 5, 6, and 7 dpi. The concentration of *Agrobacterium* cells harboring N5-Δcp-*gfp* or N5^S1-MP^-Δcp-*gfp* was 0.0001 at OD_595_. (**c**) Statistic results of GFP foci measured from N5-Δ*cp*-*gfp* and N5^S1-MP^-Δcp-*gfp* at 5, 6, and 7 dpi. All GFP foci visualized at 5 dpi were measured again by Image J at 6 and 7 dpi. The collected data were analyzed with GraphPad Prism. The dark lines indicate the mean values of GFP foci size at a certain time point.

**Figure 5 viruses-15-00157-f005:**
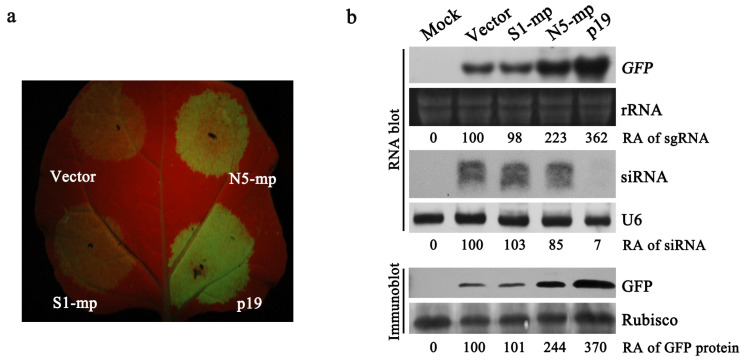
Comparison of N5 and S1 MPs in suppressing local *GFP* post-transcriptional gene silencing. (**a**) Observation of GFP fluorescence under UV light at 3 dpi. The leaves of *N. benthamiana* were co-infiltrated with *Agrobacterium* cells containing 35S:GFP and 35S:dsFP (GFP-derived inverted repeat RNA), together with the cells carrying a vector control, 35S:N5-MP, 35:S1-MP or 35S:p19. (**b**) Analyses of GFP protein, GFP transcripts and GFP-derived small-interfering RNAs (siRNAs) using Western blot and Northern blot, respectively. All these RNAs were detected by DIG-labeled DNA oligonucleotides. Ethidium bromide-stained ribosomal RNAs (rRNAs) were used as RNA loading control for analysis of GFP transcripts. U6 RNA was detected as a loading control. Rubisco protein stained by Ponceau S was used as a loading control. Relative accumulation levels of GFP transcript, GFP-derived siRNA, and GFP protein are shown below.

**Figure 6 viruses-15-00157-f006:**
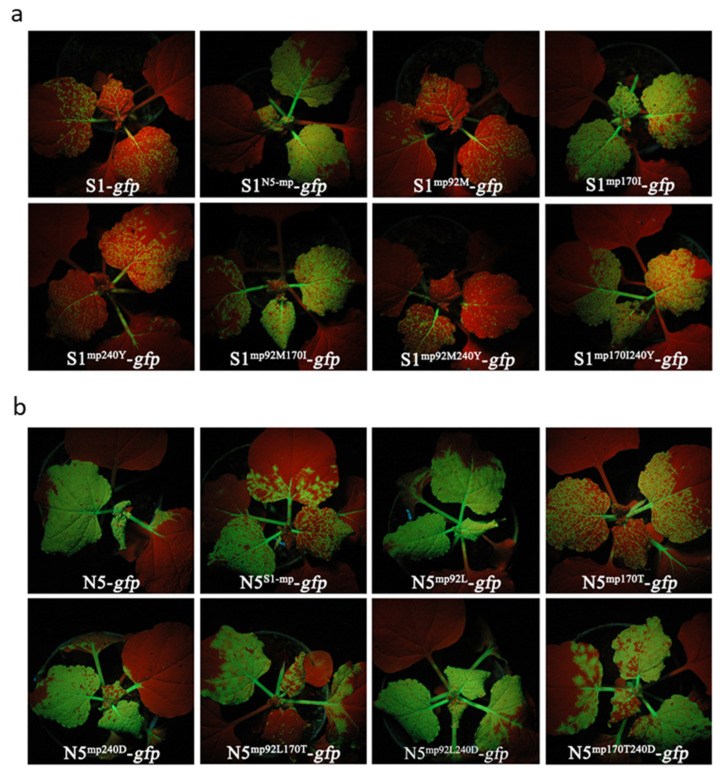
Differential green fluorescence intensities produced by S1-*gfp* and its MP mutants (**a**), or N5-*gfp* and its MP mutants (**b**) in *Nicotiana benthamiana*. The fifth true leaves were inoculated with viruses via agroinfiltration. Plants were photographed under UV light at 8 dpi.

**Figure 7 viruses-15-00157-f007:**
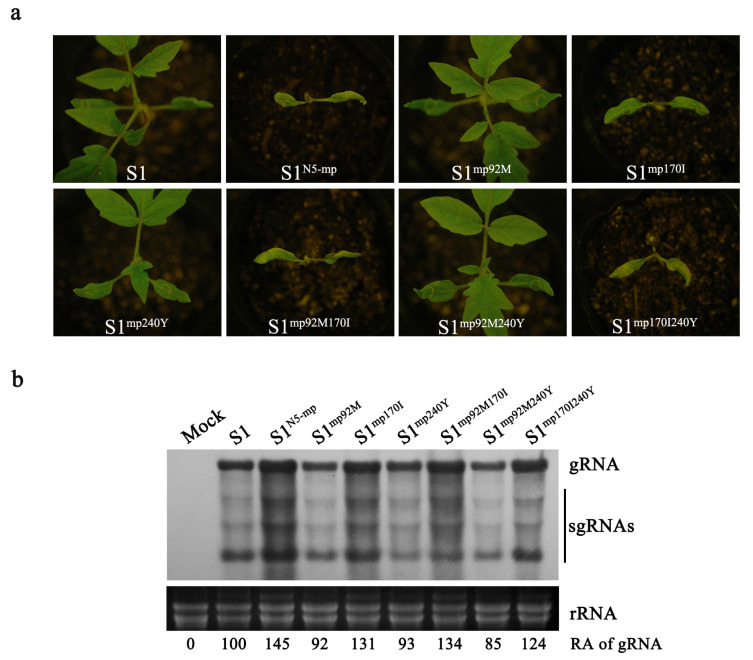
The S1 mutants harboring isoleucine at position 170 (170I) in its MP induced systemic necrosis in tomato plants. (**a**) Disease symptoms in tomato plants infected with S1 or its MP mutants, as well as S1^N5-mp^, at 7 dpi. (**b**) Steady-stage levels of viral RNAs in the upper systemic leaves were analyzed using Northern blot using a DIG-labeled DNA oligonucleotide. The bands corresponding to genomic RNA and subgenomic RNAs of these viruses were indicated with gRNA and sgRNA, respectively. Ethidium bromide-stained ribosomal RNAs (rRNAs) were used to assess relative loading amounts of RNA samples. Relative accumulation levels of gRNA are shown below.

## Data Availability

Not applicable.

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
