# Peer review of "Movement Protein Mediates Systemic Necrosis in Tomato Plants with Infection of Tomato Mosaic Virus"

_viruses, 2023, doi:10.3390/v15010157_

Round 1

Reviewer 1 Report

Dear Authors,

At the outset, I would like to point out that the work is extremely interesting. The aspects discussed in it significantly enrich the knowledge about both the tomato mosaic virus and the mechanisms of PTGS. This paper indicates a potential pathogenicity determinant in the necrotic strain of ToMV-N5 and its contribution as a weak viral suppressor. The work is written correctly, the research was done reliably and with scientific ethics.

The manuscript contains minor stylistic and editorial errors, which, however, do not affect the substantive aspect of the research carried out. Let me highlight my comments below.

Affiliations - please combine these affiliations into one and provide an e-mail address only to the corresponding author;

Introduction: 

Line 42 - please check the spelling of proteins and genes, and in this case correct to lowercase the word "Movement";

Line 57 - please make sure of the spelling style throughout the manuscript. Here, please correct the italics of the word "and" and make sure that all abbreviations are correctly described when they first appear (in this case, Solanum melogena did not appear earlier in the article);

Line 67 - Please correct the tomato torrado virus abbreviation (from TTV to ToTV);

Line 68 - If you describe virus abbreviations everywhere, apply this also to the parietaria mottle virus;

Line 73 - please expand the abbreviation to Nicotiana ;

Materials and Methods:

Line 85 -  please correctly name N. benthamiana, as well as write it in italics (similar note for lines: 133 and 134);

Line 93 - all kits and reagents should be described together with the company name and its city/state, please check it throughout the manuscript and add where it is required;

 Line 137 - correct to "TRIzol Reagent" and add all required information in brackets;

Results:

Line 159 - "DNA fragment of viral genome [...]" - mental shortcut, please reword;

In the results, please set the figures under the paragraph they relate to. In this case, figure 1 should be in line 196. Please correct this throughout the manuscript (applies to figures 2, 6 and 7);

The last sentence in each paragraph described in the results section should be included in the discussion. In a given paragraph, the results are presented, while their exact interpretation has its place in the discussion section. Please correct this throughout the "Results" paragraph. 

Discussion:

Line 377 - correct the citation to "Wieczorek and ObrÄ™palska-StÄ™plowska [45]"

Lines 395 and 396Please expand the shortcuts Nicotiana and Arabidopsis;

Lines 411-413 -  You have not studied protein interactions in this paper, and in this case, it is a great deal of uncertainty. Please correct this.

Lines 422-423note as above, you also did not investigate changes in the phosphorylation of the MP protein of ToMV. Please correct.

In the discussion, you describe the role of the MP protein as a potential weak suppressor of the ToMV virus. However, you do not mention another weak suppressor, which is the P126 protein in the case of TMV. I also miss in your analysis the comparisons to this well-known suppressor, which could significantly enrich the results obtained by you.

At the end of this paragraph, I lack a final conclusion for your research. Please complete with a 2-3 sentence summary.

Yours faithfully,

The reviewer

Reviewer 2 Report

The paper is overall of good quality and definitely of scientific merit, and contains new data on virus transport, and on the role of viral movement protein in the process of necrosis development and RNA silencing suppression.

The text, however, requires many corrections and general polishing/English editing (with some examples and suggestions indicated below – THE LIST BY FAR IS NOT EXHAUSTIVE!).

The ‘Materials and methods’ would benefit from more simplified and organized structure, especially when it comes to plasmids and clones, sometimes it’s not easy to follow.

The ‘Results’ and ‘Discussion ’sections generally are well written. However, I recommend joining sections 3.2 and 3.4 as they both deal with N5 MP-induced increased virus accumulation. Also, I would welcome more considerations and comparisons with other published data in the discussion section when it comes to RNAi suppression.

I couldn’t find Supplementary table 1.

Figure 1: green color is in the text but it appears to be grey on the picture.

In Figure 6, (a) and (b) are mistakenly placed (should be vice versa for N5 and S1).

Figures 6 and 7 are shown in the paper BEFORE their actual data and are not referenced properly in the text (i.e., Fig.6/7 is missing before the figures)

N5 92L lacks explanation in Figure 7.

Is there any quantitative data (i.e. direct measurements of RNA levels) in support of just qualitative results (gels) shown in Figs.1, 7 discussed in the ‘Discussion’ section?

Sections 3.1, 3.2: did you try to express N5 MP protein in plants naturally infected or agroinfiltrated by S1 to confirm that N5 MP may subdue RNA silencing and induce higher virus titers?

Below I include a list of noted mistakes etc. with suggested corrections.

The paper may be published after thorough revision.

INTRO

p.1, l.5-9: The affiliation for all authors seems to be the same, so please remove 4 unnecessary repetitions.

p.1, l.16: please change ‘…was not due to their functions in…’ to ‘…was not due to its functions in…’

p.1, l.18: please change ‘Sequence alignment of both MPs shows that three residues differ only between them’ to ‘Sequence alignment showed that both MPs differed by only three amino acid residues

p.1, l.18-19: please change ‘Experiments of viruses with mutated MPs indicated that the residue isoleucine’ to ‘Experiments with viruses having mutated MPs indicated that the isoleucine residue

p.1, l.21-22: phrase ‘…lethal necrosis caused by N5 is genetically determined by its MP protein’ needs another wording because here a trait is GENETICALLY determined by a PROTEIN.

p.1, l.34-36: virus names should not be italicized

p.1, l.42: please use small letter ‘m’ for ‘Movement protein

p.2, l.51: please rephrase ‘All viral proteins encoded by tobamoviruses have been documented in’ to ‘All viral proteins encoded by tobamoviruses have been shown to be important for…’

p.2, l.52: ‘For instances’ > ‘For instance

p.2, l.55: ‘…leading to occurrence of local lesions’ > ‘…leading to the development of local lesions

p.2, l.55: please remove italics for ‘and’ in ‘…Nicotiana sylvestris AND S. melongena..

p.2, l.55: please rephrase ‘…leading to extreme resistance’ to ‘…leading to extreme resistance in tomato

p.2, l.67-68: virus names should not be italicized

p.2, l.69: ‘refer to’ > ‘referred to

p.2, l.74-75: ‘residue Ile’ > ‘Ile residue

p.2, l.74-75: which position? ‘the residue Ile at position of MP’?

p.2, l.77: please rephrase ‘that trigger tomato systemic necrosis response’ to ‘that trigger systemic necrosis response in tomato

MATERIALS AND METHODS

p.2, l.85: please italicize N. Bethamiana

p.2, l.93: here and everywhere: please indicate countries of manufacturing for all reagents and equipment used in research

Section 2.2 (plasmid constructs): It’s difficult to follow the text. I recommend to shorten it significantly and make a table instead showcasing all these plasmids, and indicating exchanged fragments, primers used as well as the cloning sites.

p.3, l.128-129: please italicize Agrobacterium tumefaciens

p.3, l.133-134: please italicize N. Bethamiana

p.3, l.137: ‘Total RNAs’ > ‘Total RNA’ (here and below)

p.3, l.138: ‘…according to the product manual’ > ‘…according to the manufacturer’s instructions’

p.3, l.141: please change ‘per’ to ‘of’

p.3, l.146: please change ‘Supplemental’ to ‘Supplementary

p.4, l.148: please change ‘leave’ to ‘leaf’

p.4, l.149: please indicate reference for SDS PAGE

RESULTS

please rename subtitles 3.1-3.5 to PRESENT TENSE (‘contributed’ > ‘contributes’, etc.).

p.4, l.158: change ‘latent’ to ‘latent infection’

figures 1, 2, 4, 5, 7: indicate what acronyms gRNA, sgRNA, siRNA and rRNA mean

p.5, l.182, l.183: change ‘with the infection of’ to ‘infected with’

p.5, l.183: change ‘emerged’ to ‘developed’

p.5, l.192: change ‘had a weak pathogenicity and low accumulation’ to ‘had weak pathogenicity and accumulated to low levels’

p.5, l.192: change ‘expression’ to ‘development’

p.5, l.254: please rename subtitle 3.4 to ‘N5 MP may increase virus accumulation by stabilizing viral RNAs’ to indicate that it’s a suggestion (as stated in l.271, 272).

p.6, l.257: please replace the word ‘exchangers’ to another option (MP variants, etc.) with reference to respective figure (i.e., 4(a), etc.)

p.6, l.270: ‘viral accumulations’ > ‘virus accumulation

p.8, l.300: ‘viral levels’ > ‘virus levels

p.9, l.344: ‘with infection of’ > ‘infected with’

DISCUSSION

p.10, l.359: ‘property’ > ‘properties

p.10, l.361: change ‘expression’ to ‘development’

p.10, l.362: ‘non-necrosis’ > ‘non-necrotic

p.10, l.362: ‘…N5 MP has no advantages in facilitating’ > ‘…N5 MP does not facilitate virus…

p.10, l.364: ‘viral accumulation’ > ‘virus accumulation

p.10, l.372: ‘with infection of’ > ‘infected with’

p.10, l.373-374: ‘assists’ > ‘assist’

p.10, l.374: ‘virus escape’ > ‘virus to escape’

p.10, l.375: ‘in the virus infected tomato plants’ > ‘in virus-infected tomato plants

p.10, l.397: please italicize Arabidopsis
